

# Prediction of ACL injury incidence and analysis of key features in basketball players based on multi-algorithm models

Longfei Guo[1,2], Zhilei Cui[3], Wei Ping Loh[4] and Shazlin Shaharudin[1]

[1] Exercise & Sports Science Programme, School of Health Sciences, Universiti Sains Malaysia, Kelantan, Kubang Kerian, Malaysia
[2] Department of Physical Education, Jinzhong University, Jinzhong, Shanxi Province, China
[3] Institute of Sports Science, Taiyuan University of Technology, Taiyuan, Shanxi Province, China
[4] School of Mechanical Engineering, Universiti Sains Malaysia, Penang, Malaysia

## ABSTRACT

**Background**. Basketball players are a high-risk group for anterior cruciate ligament (ACL) injuries. This study aimed to identify the critical factors contributing to ACL injuries in male basketball players and evaluate the performance of machine learning (ML) algorithms in injury prediction.

**Methodology**. This study protocol was registered with International Standard Registered Clinical/soCial sTudy Number (ISRCTN) (Registration number: 18009799). A total of 104 male collegiate basketball players volunteered to participate in this study. Data on the athletes' profile, physical functions, basketball-specific skills, biomechanics, and electromyography (EMG) of seven lower limb muscles during unanticipated side-cutting maneuvers were collected. A 12-month follow-up was conducted to compare these variables between the injured ($n = 11$) and non-injured ($n = 93$) groups. Only the variables with significant differences between the groups were included in the predictive modeling.

**Results**. The performance of machine learning models in predicting ACL injury risk was assessed using the area under the curve (AUC) of the receiver operating characteristic (ROC). The AUC-ROC values ranged from 0.66 to 0.80, with the random forest algorithm achieving the highest performance (AUC-ROC = 0.80). The most influential predicting feature observed during the emergency stop phase, included a greater knee flexion moment, reduced knee flexion angle, increased backward ground reaction force, and increased activation of the vastus lateralis muscle.

**Conclusion**. The random forest model demonstrated superior predictive performance, providing valuable insights into the key risk factors associated with ACL injury among male basketball players. This study highlighted the importance of biomechanical testing based on sport-specific movements to accurately predict the ACL injury risk.

Subjects Data Mining and Machine Learning, Sports Injury, Sports Medicine
Keywords Biomechanics, Functional movements screening, Physical functions, Predictive modeling, Risk factors

Corresponding author
Shazlin Shaharudin, shazlin@usm.my

# INTRODUCTION

Anterior cruciate ligament (ACL) injuries are prevalent among basketball players, with an annual incidence rate ranging from 0.15% to 3.7% (*Moses, Orchard & Orchard, 2012*).

These injuries significantly impact athletic performance, as returning to play following ACL reconstruction typically requires approximately 10 months (*Jones et al., 2023*). Identifying and addressing the risk factors contributing to ACL injury are critical for effective prevention. Male and female athletes differ notably in anatomical structure, biomechanics, hormone levels, and neuromuscular control strategies, influencing ACL injury mechanisms (*Renstrom et al., 2008*). Females typically exhibit increased knee valgus, lower hamstring-to-quadriceps ratios (*Hewett, Myer & Ford, 2006*), and ligament laxity variations linked to menstrual cycles, elevating ACL injury risk (*Dos'Santos et al., 2023*). To eliminate gender as a confounding factor and enhance internal validity, this study focused solely on male basketball players.

Standardized movement tests, including the Landing Error Scoring System (LESS) (*Smith et al., 2011*) and the Cutting Movement Assessment Score (CMAS) (*Dos'Santos et al., 2019*), are widely used to evaluate the ACL injury risk in players. These tests primarily focus on the athlete's biomechanical risk factors, such as dynamic knee valgus (*Dos'Santos et al., 2021*) or landing technique (*Limroongreungrat et al., 2022*). However, the ecological validity of these standardized movement tests is often limited due to their inability in replicating the real-game scenarios' dynamic and unpredictable nature. To overcome these limitations, the unanticipated side-cutting maneuvers can be utilized as it accurately reflects the complex decision-making and dynamic challenges players encounter during competition (*Kim et al., 2014*). Notably, ACL injuries typically occur during high-risk actions—such as rapid directional changes or single-leg landings—when the knee experiences peak shear and torsional loads (*Krosshaug et al., 2007*). These high-risk instances are often masked when analyzing the movement as a whole, as key biomechanical variables—including ground reaction force, knee valgus moment, and flexion-extension angle—fluctuate dynamically throughout the motion (*Baumgart, Hoppe & Freiwald, 2017*). Conventional time-averaged approaches may therefore obscure phase-specific patterns, reducing model sensitivity. Segmenting the unanticipated side-cutting maneuver into distinct phases improves temporal resolution, enabling precise identification of critical injury-prone events and facilitating targeted biomechanical characterization.

Other contributing factors, such as lower limb and trunk strength deficits (*Raschner et al., 2012*), poor balance, and joint laxity (*Oshima et al., 2018*), may further complicate the prediction of ACL injury. To objectively quantify these ACL-related risk factors, several assessment tools have been widely applied. The Functional Movement Screen (FMS) identifies deficits in mobility and neuromuscular control associated with knee instability (*Cook et al., 2014*; *Kiesel, Plisky & Butler, 2011*). The Y-Balance Test (YBT) assesses dynamic balance and limb asymmetry; both linked to lower limb injury risk (*Plisky et al., 2006*). Deficits in basketball-specific physical attributes—such as strength, power, and agility—may impair neuromuscular control around the knee joint, thereby increasing the risk of ACL injury (*Hewett et al., 2005*).

Machine learning (ML) is a data-driven approach that employs algorithms and statistical models to enable automated classification or decision-making processes. Model performance is typically evaluated using metrics such as cross-validation and accuracy (*Litjens et al., 2017*). ML ability to manage complex and multidimensional datasets

makes it a powerful tool for analyzing biomechanical data and movement patterns, and subsequently, facilitating early detection and injury prevention. For instance, *Kolodziej et al. (2023)* employed a LASSO regression model to predict lower limb injury risk among 56 male elite adolescent soccer players using 3D motion analysis, postural control, and strength assessments. Their results identified knee extensor peak torque, hip transverse torque, and center of pressure (COP) sway during single-leg stance as key predictive indicators. Although the model demonstrated a limited predictive performance, with an accuracy of 0.58 and an AUC of 0.63, these findings underscored the importance of neuromuscular and biomechanical factors in injury risk assessment (*Kolodziej et al., 2023*). Similarly, another study involving 39 basketball players employed inertial sensors on leg stability, mobility, and load absorption. Risk levels were assessed using LESS, and a support vector machine (SVM) algorithm achieved an impressive predictive accuracy of 0.96, in which the primary risk factors identified were load absorption parameters and leg mobility (*Taborri et al., 2021*). Although LESS is commonly used to assess injury risk, its reliance on subjective judgment introduces bias (*Wexler et al., 2019*). In addition, LESS is based on two-dimensional video analysis, which limits its ability to capture the three-dimensional biomechanics critical to accurate assessment (*Hanzlíková & Hébert-Losier, 2020*). These methodological constraints may compromise their predictive validity. In contrast, actual diagnosis offers greater clinical and practical relevance. In this study, ACL injury status was confirmed using the Lachman test and magnetic resonance imaging (MRI)—the clinical gold standard—and used as outcome labels for ML modelling. This strategy supports a shift in sports medicine from risk screening to precise prevention.

Building on this foundation, *Xu et al. (2024)* developed a biomechanical and nonlinear musculoskeletal modeling approach to assess how ankle kinematics during single-leg landings affect lower-limb injury risk. By incorporating nonlinear short-term viscoelastic properties, the ACL model accurately captured ligament mechanics and improved load prediction fidelity. Similarly, *Xu et al. (2023)* developed a novel approach that integrates deep learning with musculoskeletal modeling to predict ACL loading. By extracting ankle kinematic features during single-leg landings before and after fatigue, they constructed and optimized Sparrow Search Algorithm (SSA)-extreme learning machine (ELM) and SSA-long short-term memory (LSTM) models, achieving high prediction accuracy ($R^2 = 0.9947$). In the context of real-time injury prediction, *Ren, Wang & Li (2024)* developed a real-time injury monitoring system that integrates deep learning and SVM, using polynomial fitting and principal component analysis (PCA) to extract motion features. The system achieved over 90% accuracy across multiple sports and an average prediction precision of 94.2%, outperforming traditional heuristic methods and demonstrating the promise of ML in injury surveillance. *Ayala et al. (2024)* proposed a logistic regression model identifying kinesiophobia and high-risk sport types as key predictors. With 90% accuracy and a recall of 99.16%, the model surpassed SVM and K-nearest neighbors (KNN), offering an effective tool for personalized injury prevention. While previous studies highlighted the potential of ML in injury prediction, several limitations persist. The clinical application of ML models was hindered by suboptimal model performance, restricted interpretability, and variability across different sports.

Insufficient sample size is a well-recognized limitation in ML studies, often resulting in overfitting and reduced generalizability. However, even a large-scale predictive study involving 880 athletes achieved an AUC-ROC of only 0.63. Moreover, attempts to address sample imbalances in the dataset failed to improve the outcomes (*Jauhiainen et al., 2022*) thus revealing a substantial gap in predictive performance relative to clinical applications. To address these limitations, the study incorporated a wider range of ACL injury-related factors tailored to basketball athletes. Biomechanical indicators were divided into distinct movement phases (emergency stop, initial acceleration, side-cutting) to evaluate the model's applicability in different scenarios. Synthetic Minority Oversampling Technique (SMOTE) and Gaussian noise techniques were used together to address class imbalance. Finally, the Gini Importance method was applied to quantify how each feature contributed to the model's predictive performance.

Therefore, the objective of this study was to assess the effectiveness of ML algorithms in predicting ACL injury among male basketball players and to identify key factors associated with these injuries. By addressing current limitations, the findings provide actionable insights to improve injury prevention strategies.

## MATERIALS & METHODS

The study protocol was approved by the Human Research Ethics Committee of Universiti Sains Malaysia (JEPeM-USM), study protocol code: USM/JEPeM/22040199 and adhered to the principles outlined in the Declaration of Helsinki. Written informed consent was obtained from all participants before their participation in the study.

### Participants

Participants were recruited through purposive sampling from collegiate basketball teams across various universities in Shanxi, China. Male participants aged 18 or above, engaging in more than 8 h of basketball-specific training weekly, with at least three years of competitive basketball experience, and with a negative Lachman's knee examination were included in this study. On the other hand, those with exercise-related or neurological disorders, recent hip or knee surgery, or trauma were excluded.

In ML models, there are no strict rules regarding sample size. The required sample size depends on the number of features, model complexity, task type, and data distribution. In the supervised learning model, more than 80 samples are required to achieve the mean average, and 100–200 samples can be of moderate size (*Figueroa et al., 2012*; *Gillain et al., 2019*). Therefore, following this guideline, the initial sample size was set at 120 individuals. However, 11 of them voluntarily withdrew during the testing period due to personal reasons (*e.g.*, scheduling conflicts), and five were excluded due to over 50% missing data. The final sample consisted of 104 participants, including 11 who sustained an ACL injury and 93 who remained non-injured. The mean age of the injured group was 21.6 years old; with their mean height of 186.4 cm, mean weight of 83.8 kg, and mean training duration of 4.0 years. Meanwhile, the mean age of the non-injured group was 20.5 years; with their mean height of 185.1 cm, mean weight of 80.0 kg, and mean training duration of 5.4 years.
## Data collection

This cohort study was conducted between September 2022 and March 2024. The baseline data were collected, and the participants were followed for 12 months to assess their injury status. The testing protocol was completed in a single day which spanned approximately 5 h, with a designated noon break to ensure adequate rest among the participants. All experiments were conducted at Taiyuan University of Technology's High-Level Sports Center and Biomechanics Laboratory. Data collection was performed independently by the research team to ensure consistency and reliability of the data.

All participants underwent a comprehensive assessment, which included evaluations of the basketball players' profile, physical functions, biomechanics, neuromuscular factors, and basketball-specific ACL injury risk factors. The participants were instructed to abstain from consuming functional beverages during the break time to avoid potential interference with the results. Participants were advised to wear appropriate sportswear and specialized athletic footwear, and to refrain from strenuous physical activity on the day before testing to ensure sufficient rest. To reduce the impact of personalized orthotics on biomechanical measurements (*Moyer et al., 2015*), participants were asked to discontinue their use 48 h before testing. For those with habitual knee brace use, standardized elastic braces were provided to ensure protocol consistency.

### Basketball players' profile and physical functions

Basketball players' profile, including height, weight, age, level of play, playing position, and self-reported injury history were documented for all participants. Physical functions were evaluated using the FMS and YBT (*Gil-Martín et al., 2021*). These assessments were conducted under the supervision of a certified FMS instructor, adhering to the standard movement protocols and safety precautions. The FMS comprised seven movement tasks designed to assess joint flexibility and detect asymmetries (*Triplett et al., 2021*). Each task was scored on a scale from 0 (poor) to 3 (perfect execution), with higher scores indicating superior movement quality. The dynamic balance was evaluated through YBT by measuring reach distances in the anterior, posteromedial, and posterolateral directions. Reach distances were normalized to leg length, where higher scores reflected better dynamic stability and control (*Schwiertz et al., 2020*).

### Unanticipated side-cutting maneuvers

In the unanticipated side-cutting maneuvers, the participants were required to run at a minimum speed of 3.5 ± 0.2 m/s and then to step on a force plate (*Kim et al., 2014*). Simultaneously, a light was randomly illuminated to indicate direction of either left or right, prompting the participants to respond immediately upon the light indication. Using a cross-step cutting maneuver, they then quickly stepped at a 45° angle in the indicated direction and stepped onto a second force plate before continuing to run forward to exit the test area. Participants must maintain movement continuity from the run-up through task completion, without significant pauses or unnecessary deceleration, to ensure valid test results. Prior to testing, participants completed standardized warm-up and practice sessions. Six valid trials per participant (*i.e.,* three valid trials for each direction) were included in the analysis, with a mandatory 2-minute rest between trials to prevent fatigue.

Kinematic and kinetic data during the unanticipated side-cutting movement were captured using the Vicon motion capture infrared cameras (Vantage5, Oxford Metrics Limited, Oxford, UK; 100 Hz) and Kistler 3D force plates (model 9287C, Measurement Technology, Winterthur, Zurich, Switzerland, 1,000 Hz). Wireless surface electromyography (EMG) electrodes (DTS, Noraxon, Scottsdale, AZ, USA) were affixed to the lower limb to record muscle activity from the rectus femoris, vastus medialis, vastus lateralis, long head of biceps femoris, short head of biceps femoris, medial head of gastrocnemius, and lateral head of gastrocnemius. These muscles were selected due to their significant activation during sudden cutting movements, which were closely associated with the ACL loading (*Li et al., 2014*).

A previous study identified muscle activation in the dominant leg as a primary contributor to lower limb injuries (*Svensson et al., 2018*). Thus, this study exclusively measured the muscle activation of the dominant leg, which was determined based on the participant's preferred kicking leg (*Liu et al., 2014*). Baseline muscle activity levels were established using the manual maximum voluntary contraction (MVC) test. Each muscle group performed two MVC trials that lasted approximately 5 s, with a minimum 30-second rest interval between trials. The electrode placement and movement tests were adhered to the surface electromyography (EMG) for non-invasive assessment of muscles (SENIAM) standard protocol (*Hermens et al., 2000*), as detailed in Supplemental Information 1. In order to minimize errors, only one researcher was responsible for placing the reflective markers and EMG electrodes for all participants.

### Basketball-specific skills test

After completing the biomechanical tests, participants were required to rest for over 2 h to ensure they had sufficient energy for the basketball-specific tests. The participants were instructed to refrain from eating or drinking for 30 min before the tests to minimize the potential influence of food and drinks on their performance.

The warm-up and stretching protocol was designed to match the testing sequence, aiming to activate the neuromuscular system without inducing fatigue. Based on prior studies (*Carmo et al., 2023*; *Guo, Li & Wu, 2018*), participants performed 5 min of cycling at 50 W, followed by 5 min of static stretching. Then, they completed 30-second passive stretches targeting key lower-limb (quadriceps, hamstrings, gastrocnemius, gluteus maximus), lumbar, and upper-body muscles, all under researchers' supervision. In order to ensure adequate recovery and maintain optimal performance of the participants, a 30-minute rest interval was provided between each skill test. Subsequently, the participants then performed a series of tests in the following sequence: dominant leg hop (DLH) (*Guo, Li & Wu, 2018*), countermovement jump (CMJ), squat jump (SJ), drop jump (DJ), one-repetition maximum (1-RM) weighted squat, deadlift (*Dias et al., 2005*), and the lane agility test (*Milan et al., 2019*).

### ACL injury diagnosis

Following the baseline assessments, ACL injury incidence among participants were meticulously monitored through monthly communications with the team physicians and self-reports from the players. A detailed diagnosis and description of the injury was

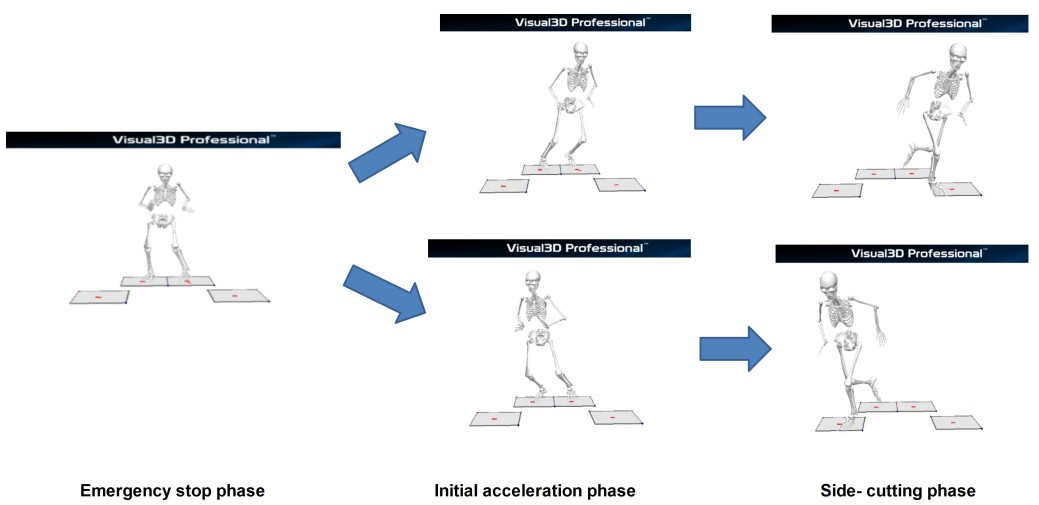

**Figure 1** **Phases of unanticipated side-cutting test.**

provided to the researcher by an orthopedic specialist upon the occurrence of the injury. All participants diagnosed with an ACL injury exhibited positive Lachman test results, and the severity of the injuries was further confirmed through MRI. As of March 31, 2024, a total of 11 ACL injury cases had been confirmed among the participants.

## Data analysis
### Biomechanical data analysis

Based on a previous study (*Li et al., 2014*), biomechanical data were collected and analyzed during three key phases of the unanticipated side-cutting maneuvers, namely the emergency stop (ES), initial acceleration (IA), and side-cutting (SC) phases, as illustrated in Fig. 1.

Firstly, the ES phase began when the participants initiated the test and progressed until both feet contacted the force platform during their braking motion where they adjusted their center of gravity (COG) based on the left or right directional cues. Next, in the IA phase, the participants adjusted their COG in response to the indicator lights. Then, the participants initiated a push-off motion toward the indicated direction while keeping both feet grounded. In both phases (ES and IA), kinetic and kinematic variables were extracted at the timepoint whereby the peak ground reaction force (GRF) was observed during double-leg stance. The extracted data included GRF peaks, center of pressure (COP), joint forces and moments at the hip, knee, and ankle, as well as surface EMG signals from seven lower-limb muscles. The moments were calculated as external moments using inverse dynamics, based on kinematic data and ground reaction forces.

Lastly, in the SC phase, following the push-off, the participants executed a side-cutting maneuvers. As the COG shifted forward, they landed on the force platform with one leg. In order to ensure consistency in movement, all participants performed the side-cutting maneuvers using a crossover step at this phase. Also, as this phase involved a single-leg landing, only biomechanical and EMG data from the supporting leg were included in the analysis for this phase. Specifically, these included GRF peaks, joint forces and moments

at the hip, knee, and ankle, as well as surface EMG signals from seven muscles of the supporting leg.

Meanwhile, the biomechanical assessment was analyzed in three-dimensional (3D) plane with three axes: $X$-axis for flexion-extension, $Y$-axis for adduction-abduction, and $Z$-axis for internal-external rotation. The kinematic and kinetic data of the ankle, hip, and knee joints were adhered to the right-hand rule to ensure a standardized representation of data (*Jandacka et al., 2018*). In addition, the GRF positive values were defined as upward, rightward, and forward directions.

Among the 104 participants, 87 were right-leg dominant and 17 were left-leg dominant. Notably, all 11 injured participants were right-leg dominant. Hence, data from the 17 left-leg dominant participants were converted to align with the dominant leg and non-dominant leg categories to standardize the analysis for consistency in comparing the muscle activation and biomechanical data.

Force plates and surface EMG devices were integrated into the Vicon Nexus system (Version 2.8.1, Oxford Metrics Limited, Oxford, UK). Data synchronization acquisition was achieved using the Nexus plugins to ensure temporal consistency across the collected data. Following that, the integrated system exported data from the force plate, motion capture, and EMG into a C3D file format. Subsequent analysis of the C3D files was performed using Visual 3D (Version 6.1.2, C-Motion, Inc., Germantown, MD, USA). Specifically, Hanavan's 15-link multi-rigid body model, recognized as one of the most complex and accurate representations of the human body, was constructed in Visual 3D (*Robertson, 2013*). The data underwent filtering with a 4th-order Butterworth filter, where the cutoff frequency for kinematic and kinetic data was 8 Hz and 30 Hz (*Weeks, 2023*), respectively. The EMG data were then processed with high-pass filter with a cutoff frequency of 50 Hz, full-wave rectification, and low-pass filter with a cutoff frequency of 6 Hz (*Merletti & Di Torino, 1999*). Finally, all the processed data were exported in CSV format for further analysis.

### Descriptive statistics

The risk factors associated with ACL injury among participants were categorized into three distinct themes, namely the athlete's profile and physical functions, biomechanical factors during unanticipated side-cutting maneuvers, and basketball-specific skills. The data were further grouped into two primary groups based on the participants' injury status, which were injured and non-injured groups. The distribution of numerical variables was assessed using the Shapiro–Wilk test for normality. Variables conforming to normal distribution were summarized as mean $\pm$ standard deviation (SD) and analyzed using independent $t$-tests, with Cohen's d utilized to quantify effect size (0.20, 0.50, and 0.80 denoting small, medium, and large effects, respectively) (*Cohen, 1988*). Variables deviating from normality were presented as median (interquartile range, IQR) and analyzed using Mann–Whitney U tests, with effect sizes represented by rank biserial correlation (range: −1 to +1) (*Cureton, 1956*). Categorical variables were compared using Chi-squared tests, with Cramér's V coefficient (range: 0 to 1) indicating effect size (*McHugh, 2013*). The numerical values of these effect sizes reflect the magnitude of differences or associations between groups,

enhancing interpretation of statistical significance. Only factors with statistical significance of $p < 0.05$ were included in the subsequent ML model construction, as summarized in Table 1. All statistical analyses were conducted using SPSS version 27.0 (IBM, Armonk, NY, USA).

## Machine learning

### Data pre-processing

Using the interquartile range (IQR) method, 399 outliers (2.12% of the original dataset) were removed (*Vinutha, Poornima & Sagar, 2018*). These outliers primarily resulted from marker-tracking errors (*e.g.*, trajectory deviations during high-speed movements) and biomechanically implausible data (*e.g.*, joint angles exceeding physiological ranges). During the crossover step of side- cutting, ankle joint kinematic data loss occurred due to marker occlusion, a common issue in dynamic movements. Missing data (1.21% of the total data in this phase) were imputed using Lagrange interpolation (*Xiong, Guo & Wu, 2021*). Trials with over 50% missing data were excluded to prevent excessive interpolation from compromising biomechanical validity. The min-max scaling was applied to standardize the dataset and mitigate the impact of differing units and dimensions among features. The variance inflation factor (VIF) was employed to identify multicollinearity to enhance model stability. The data was bootstrapped randomly 100 times to determine the highest VIF value for each feature. Features with VIF values exceeding 5, indicating significant multicollinearity, were excluded from the dataset (*O'brien, 2007*).

### Data imbalance handling

The original dataset exhibited an imbalance in data collection and labeling. For instance, categories such as basketball-specific qualities and physical characteristics contained only a single entry per category, while the biomechanical data included multiple entries per category. Thus, minority samples were oversampled and combined with the majority class to address this imbalance. Consequently, the dataset used for modeling comprised injured samples ($n = 65$) and non-injured samples ($n = 529$). In order to enhance the dataset robustness, Gaussian noise with a mean of 1 and an SD of 0.1 was added. Each perturbation employed unique random values to simulate the real-world noise. By introducing the perturbation, realistic variations were simulated, which improved the dataset's generalizability and reliability (*Ye et al., 2023*).

The augmented dataset was divided into a training set (90%) and a test set (10%) to ensure the model had sufficient samples for learning complex patterns within the data (*Miawarni et al., 2022*). A stratified 10-fold cross-validation was then used to further enhance the model performance by maintaining the consistency of class proportions with the entire dataset. This approach was particularly crucial for dataset with small samples and could be an effective parameter tuning (*Wang, 2011*). Additionally, the SMOTE was applied to oversample the injured group to 1,285 samples within each training subset during the cross-validation. As a result, the data balance was achieved. The overall data handling process is illustrated in Fig. 2.

After augmenting the dataset, Vapnik–Chervonenkis (VC) dimension analysis was conducted to evaluate the sample requirements and to ensure model compatibility. With

**Table 1 Descriptive statistics for all athletes on the significantly different indicators according to injury status (injured/uninjured).**

| Factors | Injured players (n = 11) | Non-injured players(n = 93) | S-Wilk (P)/RR | P | Effect size |
|---|---|---|---|---|---|
| **Basketball players' profile and physical function (4)** | | | | | |
| YBT - right leg combined score | 0.807 ± 0.082 | 0.947 ± 0.124 | 0.498/0.082 | T-test: 0.000 | 1.333 |
| ③YBT - left leg combined score | 0.8297 (0.7309, 0.9143) | 0.9241 (0.8453, 1.0454) | 0.083/0.007 | U-test: 0.005 | 0.888 |
| Injury history | Injury History: 7 No Injury History: 4 | Injury History: 27 No Injury History: 66 | RR = 3.603 | $X^2$-test: 0.048 | 0.186 |
| ④Weekly basketball hours | >15h: 9; ≤15h: 2 | >15h: 41; ≤15h: 52 | RR = 4.860 | $X^2$-test: 0.018 | 0.212 |
| **Basketball-Specific Quality (2)** | | | | | |
| Relative deadlift | 1.380 ± 0.247 | 1.56 ± 0.278 | 0.865/0.074 | T-test: 0.042 | 1.042 |
| ⑦Squat jump | 46.363 ± 8.237 | 53.032 ± 9.798 | 0.198/0.050 | T-test: 0.033 | 0.752 |
| **Biomechanical Factors (emergency stop phase) (17)** | | | | | |
| ⑧FP4(y) | −304.61 (−474.569, −134.775) | −202.87 (−405.608, −65.067) | 0.000/0.000 | U-test: 0.016 | 0.159 |
| R- cop-distance (X) | 0.045 (0.037, 0.061) | 0.059 (0.044, 0.076) | 0.070/0.000 | U-test: 0.001 | 0.267 |
| R Shank angle(Y) | 3.228 (−4.326, 10.796) | 5.477 (−1.236, 12.739) | 0.592/0.000 | U-test: 0.033 | 0.169 |
| L-ankle dorsiflexion angle | 7.123 (5.801, 28.769) | 14.222 (3.756, 25.419) | 0.080/0.000 | U-test: 0.036 | 0.156 |
| L- hip Adduction angle | 22.377 (13.505, 26.612) | 15.765 (9.49, 22.671) | 0.541/0.008 | U-test: 0.000 | 0.282 |
| L-hip Extension Moment | −3.823 (−8.026, −0.26) | −0.654 (−5.165, −0.044) | 0.000/0.000 | U-test: 0.005 | 0.218 |
| ⑩L-knee Flexion angle | −58.865 (−69.649, −47.17) | −65.049 (−74.327, −55.249) | 0.102/0.00 | U-test: 0.001 | 0.331 |
| L- knee Internal rotation angle | −21.417 (−38.827, −9.281) | −15.821 (−31.318, −3.91) | 0.001/0.000 | U-test: 0.038 | 0.192 |
| ① L-knee Flexion Moment | −1.393 (−2.663, −0.175) | −0.239 (−1.609, −0.099) | 0.000/0.000 | U-test: 0.000 | 0.267 |
| L-knee Internal rotation Moment | −0.117 (−1.415, −0.014) | −0.022 (−0.167, 0.069) | 0.000/0.000 | U-test: 0.000 | 0.163 |
| R-ankle dorsiflexion angle | 20.627 (9.511, 33.735) | 16.119 (6.08, 25.098) | 0.000/0.000 | U-test: 0.028 | 0.148 |
| R- ankle Eversion angle | −1.503 (−25.907, −7.297) | −0.698 (−19.1, −14.759) | 0.001/0.001 | U-test: 0.048 | 0.135 |
| R- hip Internal rotation angle | 9.873 (4.225, 14.988) | 6.507 (0.601, 13.846) | 0.000/0.000 | U-test: 0.028 | 0.220 |
| ⑤ R- knee Flexion angle | −60.538 (−70.299, −52.883) | −63 (−75.27, −56.387) | 0.159/0.001 | U-test: 0.014 | 0.160 |
| Activation level of the rectus femoris | 0.25 (0.139, 0.497) | 0.155 (0.043, 0.442) | 0.002/0.000 | U-test: 0.008 | 0.205 |
| ⑪Activation level of vastus lateralis | 0.242 (0.119, 0.489) | 0.168 (0.064, 0.386) | 0.000/0.000 | U-test: 0.006 | 0.193 |
| Activation level of biceps femoris short head | 0.279 (0.08, 0.554) | 0.175 (0.057, 0.459) | 0.002/0.000 | U-test: 0.050 | 0.195 |
| **Biomechanical factors (Initial acceleration- left direction) (12)** | | | | | |
| FP3(x) | −370.82 (−544.271, −208.661) | −258.273 (−419.043, −82.33) | 0.007/0.000 | U-test: 0.049 | 0.205 |
| R- cop-distance(X) | 0.048 (0.035, 0.055) | 0.054 (0.045, 0.07) | 0.225/0.000 | U-test: 0.003 | 0.540 |
| R-Tibia -angle(Y) | 9.317 (4.867, 14.637) | 12.286 (7.013, 17.032) | 0.070/0.000 | U-test: 0.047 | 0.148 |
| L-ankle plantarflexion Moment | −0.528 (−2.111, −0.012) | −0.009 (−0.66, −0.002) | 0.009/0.000 | U-test: 0.007 | 0.489 |
| L- hip Abduction angle | 23.609 (12.369, 27.406) | 15.759 (9.727, 23.076) | 0.153/0.007 | U-test: 0.012 | 0.279 |
| L-hip Flexion Moment | −3.081 (−7.048, −0.144) | −0.465 (−3.283, −0.033) | 0.013/0.000 | U-test: 0.005 | 0.235 |
| L- hip Abduction Moment | 0.324 (0.132, 1.03) | 0.08 (0.025, 0.354) | 0.006/0.000 | U-test: 0.011 | 0.188 |
| L- knee Flexion Moment | −1.52 (−2.922, −0.113) | −0.208 (−0.919, −0.083) | 0.004/0.000 | U-test: 0.002 | 0.185 |
| ②L- knee Internal rotation Moment | −0.258 (−1.102, −0.023) | −0.049 (−0.242, −0.029) | 0.004/0.000 | U-test: 0.002 | 0.236 |
| R- ankle Inversion angle | 11.65 ± 26.00 | 0.44 ± 28.92 | 0.057/0.066 | T-test: 0.025 | 0.204 |
| R- ankle Eversion Moment | −0.288 (−0.479, −0.105) | −0.402 (−0.619, −0.177) | 0.064/0.004 | U-test: 0.023 | 0.238 |
| R- hip External rotation Moment | −0.318 (−0.605, −0.115) | −0.467 (−0.679, −0.264) | 0.012/0.000 | U-test: 0.041 | 0.307 |

**Table 1** (*continued*)

| Factors | Injured players (n = 11) | Non-injured players(n = 93) | S-Wilk (P)/RR | P | Effect size |
|---|---|---|---|---|---|
| | **Biomechanical Factors (Initial acceleration- right direction) (3)** | | | | |
| L- hip Flexion angle | 80.332 (72.192, 93.226) | 73.924 (63.132, 84.599) | 0.050/0.000 | U-test: 0.018 | 0.333 |
| L- knee Internal rotation Moment | −0.091 (−0.566, −0.043) | −0.035 (−0.132, −0.032) | 0.013/0.000 | U-test: 0.004 | 0.083 |
| R- ankle Eversion Moment | −0.818 (−1.275, −0.365) | −0.489 (−0.852, −0.265) | 0.734/0.000 | U-test: 0.026 | 0.035 |
| | **Biomechanical Factors (Side-cutting phase- left direction) (8)** | | | | |
| ⑨Tibia- angle (X) | 32.37 (23.98, 42.10) | 39.44 (29.29, 46.85) | 0.382/0.000 | U-test: 0.019 | 0.259 |
| Ankle External rotation angle | −1.16 (−13.19, −2.04) | −0.83 (−4.38, −8.63) | 0.011/0.000 | U-test: 0.013 | 0.235 |
| Ankle dorsiflexion Moment | 0.02 (0.01, 0.03) | 0.02 (0.01, 0.02) | 0.000/0.000 | U-test: 0.043 | 0.307 |
| ⑥Hip Flexion angle | 54.94 (40.44, 66.30) | 63.15 (51.74, 78.12) | 0.000/0.000 | U-test:0.014 | 0.483 |
| Knee External rotation angle | −21.70 (−26.15, −13.87) | −13.64 (−26.54, −0.58) | 0.747/0.000 | U-test: 0.032 | 0.340 |
| Knee Flexion Moment | −0.08 (−0.13, −0.05) | −0.12 (−0.17, −0.06) | 0.000/0.747 | U-test: 0.031 | 0.529 |
| Knee Adduction Moment | 0.02 (0.02, 0.05) | 0.04 (0.01, 0.07) | 0.000/0.000 | U-test: 0.044 | 0.328 |
| Knee External rotation Moment | −0.01 (−0.03, −0.01) | −0.04 (−0.06, −0.00) | 0.000/0.000 | U-test: 0.044 | 0.610 |
| | **Biomechanical Factors (Side-cutting phase- right direction) (4)** | | | | |
| Ankle Eversion angle | 23.54 ± 23.13 | 9.69 ± 23.01 | 0.000/0.000 | T-test: 0.002 | 0.602 |
| Ankle Eversion Moment | 0.01 (0.01, 0.02) | 0.01 (0.01, 0.01) | 0.000/0.000 | U-test: 0.005 | 0.320 |
| Knee Flexion angle | −47.33 (−54.83, −36.36) | −52.11 (−56.74, −47.62) | 0.000/0.000 | U-test: 0.023 | 0.256 |
| Knee Flexion Moment | −0.06 (−0.10, −0.02) | −0.09 (−0.13, −0.05) | 0.000/0.000 | U-test: 0.022 | 0.262 |

**Notes.**
FP3(x): Ground reaction forces in the lateral direction on Force Platform 3.
FP4(Y): Ground reaction forces in the anterior-posterior direction on Force Platform 4.
Tibia -angle (X): Angle of the tibia of the leg relative to the ground in the sagittal plane; + values indicate Forward.
Tibia -angle (Y): Angle of the leg tibia in the frontal plane relative to the ground; + values indicate to the left.
R- cop-distance (X): Lateral distance from the center of ground reaction pressure to the ankle joint.
**a:** For non-normally distributed variables [Md (Q1, Q3)], the Mann–Whitney U test was applied.
**b:** For normally distributed variables ($\bar{X}$ ± S), the *t*-test was used.
**C:** For categorical variables Perform chi-square test.
①, ②, ③, ④, ⑤, ⑥, ⑦, ⑧, ⑨, ⑩, ⑪: Machine learning model output feature sorting.

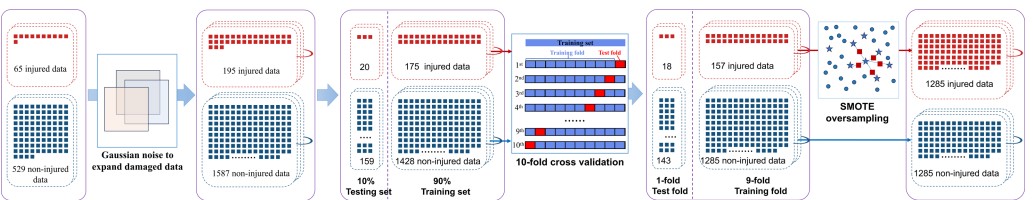

**Figure 2** Schematic diagram of processing imbalanced data.

that, the risk of overfitting was mitigated (*Chen et al., 2019*). The sample demand equation used for this purpose is listed below:

$$N \geq \frac{VC \cdot log\left(\frac{1}{\delta}\right) + log\left(\frac{1}{\epsilon}\right)}{\epsilon}$$

where, $N$ = the minimum required sample size; $VC$ = the complexity of the model; $\delta$ = the complement of the confidence level ($\delta = 0.1$); $\epsilon$ = the generalization error ($\epsilon = 0.1$).

The VC dimension of various algorithms was influenced by several factors, such as the number of features, the radial basis function (RBF) kernel, the number of decision trees (n_estimators), and the maximum depth (max_depth). The final optimized parameters are

shown in Supplemental Information 2. In this study, a total of 45 features were included in the model. Based on the number of features in the model and the total sample size after the SMOTE augmentation, the model complexity was managed by adjusting the number of decision trees and maximum depth. Cross-validation was subsequently employed to identify the optimal parameter configuration to ensure stable performance on the validation set.

### Choice of classifiers

Four widely used algorithms, namely the support vector machines (SVM) (*Surasak, Praking & Kitchat, 2023*), random forest (RF) (*Briand et al., 2022*), XGBoost (*Priscilla & Prabha D. Technology, 2020*), and logistic regression (*Stylianou et al., 2015*) were selected as classifiers for the injury prediction model in this study. Each algorithm offered unique classification and prediction strengths, enabling a comprehensive evaluation of their performance in predicting ACL injury among the participants. The models were compared using metrics such as the area under the receiver operating characteristic curve (AUC-ROC), accuracy, precision, recall, F1-score, sensitivity, and specificity. Among these, AUC-ROC provides a comprehensive measure of a classification model's discriminative ability across different thresholds (*Li, 2024*). Therefore, it was selected as the primary evaluation metric. The AUC-ROC values were classified into different categories, such as excellent (0.90–1.00), good (0.80–0.89), fair (0.70–0.79), poor (0.60–0.69), and fail (0.50–0.59) (*Krosshaug et al., 2016*). Higher values across all metrics generally indicate better model performance. All analyses were conducted using PyCharm Professional (Version 2024.1.4, JetBrains, Prague, Czech Republic).

## RESULTS

A total of 50 significantly different features were identified across three groups of risk factors associated with ACL injury among the participants, namely the basketball players' profile and physical functions (four indicators), basketball-specific qualities (two indicators), and biomechanical factors (44 indicators). Following VIF screening, five features with moderate multicollinearity were excluded namely left hip adduction angle (5.70) and right ankle eversion angle (5.54) during the ES phase, and right ankle inversion angle (6.95), left hip flexion moment (5.29), and left hip flexion abduction moment (5.67) during the IA phase. A total of 45 features were retained for ML modeling.

SVM, RF, logistic regression, and XGBoost algorithms were utilized to develop predictive models, trained on the training set and validated on an independent test set. To ensure robustness, each algorithm was executed 10 times in a loop. The performance metrics were calculated for each algorithm, including the AUC-ROC, accuracy, precision, recall, F1-score, sensitivity, and specificity. These results are summarized in Table 2, with RF demonstrating the most superior performance. For instance, RF achieved the highest values for accuracy ($0.9619 \pm 0.0376$), precision ($0.6667 \pm 0.4714$), F1-score ($0.61330 \pm 0.4431$), and specificity ($0.9947 \pm 0.0166$).

DeLong's test was employed to statistically compare the AUC-ROC values of the four models, ensuring that observed differences were not due to random variation (*De Long,*

**Table 2 Performance of each prediction model $\bar{X} \pm \sigma$.**

|  | XGboost | Random forest | Logistic regression | SVM |
|---|---|---|---|---|
| Accuracy | 0.9421 ± 0.0307 | 0.9619 ± 0.0376 | 0.8129 ± 0.0934 | 0.6736 ± 0.0903 |
| Precision | 0.5167 ± 0.4191 | 0.6667 ± 0.4714 | 0.2769 ± 0.1518 | 0.1432 ± 0.1113 |
| F1 Score | 0.5113 ± 0.3745 | 0.6133 ± 0.4431 | 0.3761 ± 0.1774 | 0.2316 ± 0.1747 |
| Sensitivity | 0.5967 ± 0.4595 | 0.6021 ± 0.4595 | 0.7017 ± 0.3496 | 0.6542 ± 0.4743 |
| Specificity | 0.9737 ± 0.0372 | 0.9947 ± 0.0166 | 0.8216 ± 0.1119 | 0.6758 ± 0.0966 |
| AUC | 0.7868 ± 0.2174 | 0.7974 ± 0.2273 | 0.7608 ± 0.1613 | 0.6629 ± 0.2392 |

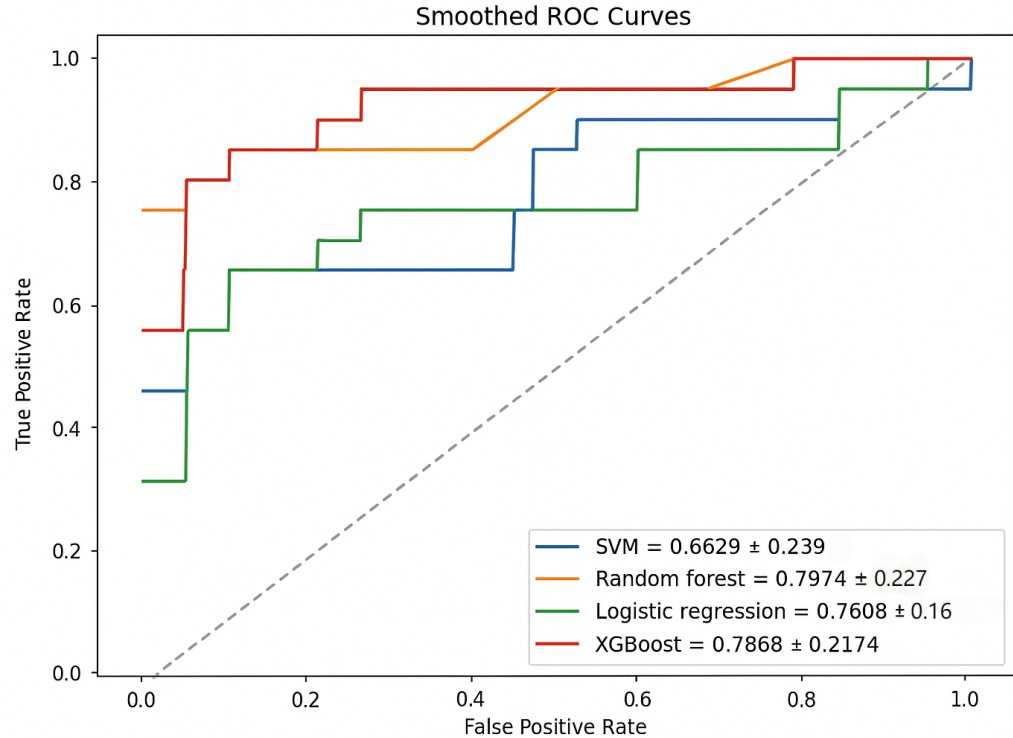

**Figure 3 AUC-ROC curves for each model.**

*De Long & Clarke-Pearson, 1988*). The results indicated no significant differences between any model pairs (RF *vs.* SVM: $p = 0.214$; RF *vs.* XGBoost: $p = 0.916$; RF *vs.* Logistic Regression: $p = 0.683$; SVM *vs.* XGBoost: $p = 0.241$; SVM *vs.* Logistic Regression: $p = 0.298$; XGBoost *vs.* Logistic Regression: $p = 0.764$). Nevertheless, RF achieved the highest AUC-ROC (0.7974), as shown in Fig. 3, and consistently outperformed the other models across most evaluation metrics.

The Gini Importance method evaluates feature's significance by calculating the total reduction in Gini impurity during node splits (with higher values indicating greater discriminative power) (*Zhang et al., 2023*). In this study, we applied this method to identify

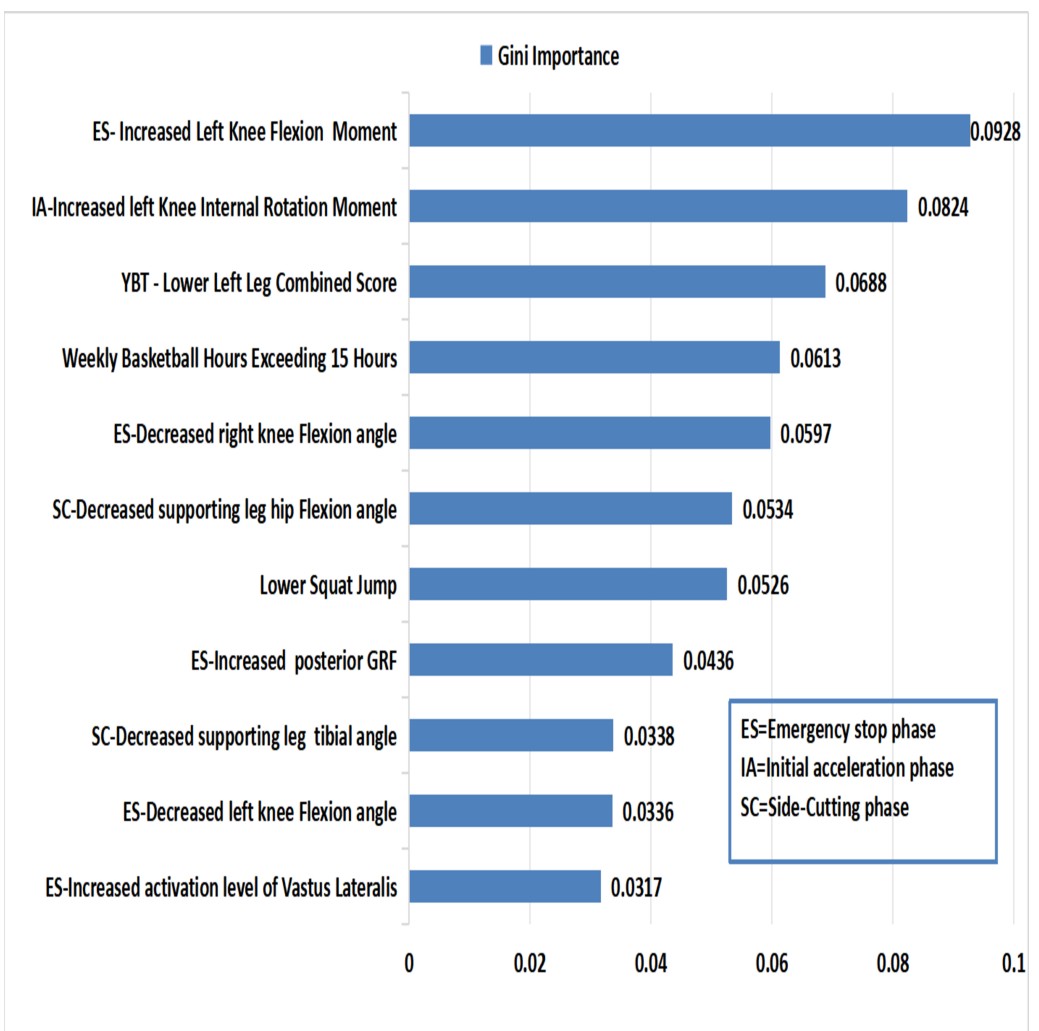

**Figure 4    Ranks of important features in random forest model.**

key predictive features (importance > 0.3) for injury classification. The top 11 features are highlighted in Fig. 4.

During the ES phase, the key predictive features for ACL injury include increased knee flexion moment, reduced knee flexion angle, elevated posterior GRF, and heightened activation of the vastus lateralis. During the IA phase, a greater knee internal rotation moment emerged as a significant risk factor. In the following SC phase, decreased hip flexion angle and tibial inclination angle contributed to the ACL injury risk. Additionally, training duration exceeding 15 h per week, a lower composite score in the Y-Balance Test (YBT), and poor SJ performance are associated with an increased risk of ACL injury.

## DISCUSSION

The present study aimed to predict ACL injury in male basketball players by analyzing the multidimensional risk factors using ML models. Physical, biomechanical, and basketball-specific data were collected and monitored over 12 months for ACL injury occurrence. These datasets were analyzed using ML algorithms to determine the most effective model for ACL injury prediction among basketball players.

The findings from this study indicated that RF emerged as the most reliable algorithm for predicting ACL injury in basketball players. By examining the factors with significant weights in the RF model, valuable insights into potential risk factors for ACL injury were identified, providing a foundation for targeted prevention strategies for this population.

### Basketball players' profile and physical functions

Based on the model results within the physical function, total weekly basketball training time and the composite score of left leg in the YBT test were significant predictors for ACL injury. Data from relative risk (RR) analysis found that those who attended training for more than 15 h per week had a 4.86-fold increase in ACL injury risk, compared to those who trained less. Similar findings were reported by *Stojmenovic et al. (2017)*, in which those who trained for more than 10 h weekly had 7.54 times higher odds of ACL injury than those with shorter training durations. The association between training volume and ACL injury risk likely arises from the combined effects of cumulative mechanical loading and neuromuscular fatigue, which together amplify tissue stress and disrupt movement patterns over time (*Taylor & Burkhart, 2025*). Therefore, defining the threshold for weekly training-load is essential. Prospective studies should pair workload monitoring with systematic injury surveillance to identify athlete-specific limits and refine targeted prevention strategies. Meanwhile, the lower dynamic balance composite scores in the YBT non-dominant leg were associated with an increased risk of ACL injury. A study found that a forward reach difference greater than four cm between limbs in the YBT increased the risk of lower limb injury by 2.5 times (*Plisky et al., 2006*). Poor dynamic balance of the non-dominant leg compared to the dominant leg (*Promsri et al., 2020*) may increase its susceptibility to ACL injury. Thus, this finding highlights the importance of assessing dynamic balance deficiencies as an injury prevention strategy.

### Biomechanical factors

In the ES phase of basketball games and training, sudden stop motions during rapid running or breakthroughs are common (*Krosshaug et al., 2007*). These sudden stop motions may cause the tibia and femur to move closer to each other and the subsequent shear force posed a significant damage risk to the ACL (*Emerson, 1993*). The predictive model in this study found that larger knee flexion moment, smaller knee flexion angles, greater posterior GRF, and higher activation levels in the vastus lateralis muscles were significant predictors of ACL injury risk during the ES phase.

A larger knee flexion moment was the most significant predictive factor for ACL injury. It was demonstrated that players with ACL injury exhibit higher peaks in knee flexion moment during landing compared to uninjured players (*Leppänen et al., 2017*).

*Xie et al. (2013)* indicated that greater knee valgus angles and subsequent quadriceps activation lead to increase knee flexion moment, which further increase the ACL injury risk. In this study, the injured group exhibited a smaller knee flexion angle, compared to the non-injured group. This was in line with the findings from *Myers et al. (2011)*, who reported that hard landings had an increased risk of ACL injury with a greater GRF peak. A hard landing resulted in a significantly smaller knee flexion angle than a soft landing, which was even more vulnerable in unanticipated competitive scenarios. Another intervention study on the landing techniques found that increasing the knee flexion angle during landing significantly reduced the vertical GRF and knee flexion moment, hence, reducing the ACL injury risk (*Favre et al., 2016*). These biomechanical advantages may be particularly pronounced in experienced athletes. Our findings support this training-related effect whereby the non-injured group had a longer history of training (5.4 years) than the injured group (4.0 years). It is plausible that experienced players are able to lower their center of mass by adopting greater joint flexion angles during abrupt stopping tasks—a mechanical strategy that is likely to mitigate the risk of ACL injury.

In addition, this study also found that increased posterior GRF was an important predictor for ACL injury. During the ES phase, the posterior GRF increased rapidly, especially on the weaker leg which potentially led to an increase in ACL loading and injury risk (*Yu, Lin & Garrett, 2006*). A study was conducted by *Kim et al. (2016)* to assess the lower limb biomechanics of players during side-cutting maneuvers. It was reported that a peak posterior GRF during the unanticipated emergency stops increase the risk of ACL injury, due to an increase in the anterior shear force on the tibia. During the stop-landing, GRF transmits to the soft tissues through the tibia, while the femur moves posterior relative to the tibia. As a result, this motion stretches the ACL beyond its capacity to prevent such displacement, thus increasing the risk of ACL injury (*Boden & Sheehan, 2022*).

During the ES phase, the activation level of vastus lateralis muscle was significantly higher in the injured group. The vastus lateralis muscle plays a synergistic role in maintaining dynamic stability of the knee joint during movements (*Mellor & Hodges, 2006*). However, an imbalance between the quadriceps and hamstrings caused by the predominant contraction significantly increased the risk of ACL injury (*Myer et al., 2005*). Furthermore, excessive activation of the quadriceps during abrupt stopping and landing movements compromised the dynamic stability of the knee joint due to the diminished protective role of surrounding muscles and ligaments which heightened the likelihood of ACL injury (*Hewett, Myer & Ford, 2006*).

Notably, high external knee flexion moments can occur despite relatively modest knee flexion angles. Although this inverse relationship may appear counterintuitive, mechanical analyses of unanticipated side-cutting have shown that elevated posterior ground reaction forces during early stance substantially increase external knee loading, even in the presence of limited joint excursion (*Cortes, Onate & Van Lunen, 2011*). Additionally, insufficient preparatory muscle activation and altered neuromuscular control under unanticipated conditions may limit knee flexion while still producing high external loads (*Besier, Lloyd & Ackland, 2003*).

When the random indicator light directed movement to the left, the COG shifted leftward. This creates a greater internal rotation moment at the knee of the supporting leg. This internal rotation emerged as a significant predictive factor for ACL injury during the IA phase. For instance, the motion drove the proximal tibia to rotate internally, accompanied by dynamic knee valgus. The combination of knee valgus and internal rotation moments significantly increased the shear stress. Subsequently, the ACL injury risk was heightened when the coordinated action of the knee joint and surrounding muscles were not effectively controlled. A study by *Nedergaard et al. (2020)* supported these findings, highlighting that certain lateral cutting movements induced greater knee flexion during the push-off phase, accompanied by greater internal rotation moment and muscle pre-activation levels, contributing to ACL injury risk. Furthermore, during unanticipated lateral cutting movements, the limited time for postural adjustment resulted in insufficient muscle activation to counteract external joint loads. As a result, this inadequacy increased the internal and external rotation torques at the knee, further exacerbating the risk of ACL injury (*Besier et al., 2001*).

A reduced hip flexion angle and a reduced tibial inclination angle during the SC phase were identified as significant predictors of ACL injury. A smaller hip flexion angle predisposed the knee to hyperextension or a straight posture, which subjected the ACL to greater tensile force. Meanwhile, a hyperextended knee is particularly vulnerable to injury when exposed to shear forces (*Devita & Skelly, 1992*). Additionally, a reduced tibial inclination angle often correlated with a higher COG and a reduced knee flexion angle. This combination resulted in a hard landing, thereby increasing the risk of ACL injury (*Taylor et al., 2017*). Importantly, greater body weight exacerbated these detrimental effects. Our statistical analysis revealed that injured players demonstrated significantly higher mean body weight (83.8 kg) compared to non-injured players (80.0 kg). These findings are supported by the study of *Kaplan et al. (2020)*, which demonstrated that increased body mass during landing generates higher knee shear forces, thereby substantially elevating ACL loading.

## Basketball-specific skills factors

The squat jump (SJ) emerged as the most significant predictor of ACL injury among basketball-specific qualities, where the injured group typically scored lower on the SJ test compared to their non-injured counterparts. SJ performance was reported as a key indicator of an athlete's explosive strength and neuromuscular efficiency (*Slater & Hart, 2017*). Basketball is a sport that heavily relies on explosive power for actions such as grabbing rebounds or making defensive steals. The ability to create sufficient force in a split second determined the success in executing these high intensity actions (*Pomohaci & Sopa, 2021*).

As a component of structured plyometric training, the squat jump (SJ) may serve as an effective exercise to enhance lower limb explosive strength by increasing muscle contraction speed during take-off and landing phases, thereby contributing to neuromuscular adaptations that may protect the ACL (*Malisoux et al., 2006*). Furthermore, a high SJ score reflected the superior muscle co-activation efficiency of the players. This exercise involved

eccentric contraction during the squatting phase and concentric contraction during the jumping phase, primarily targeting the quadriceps, gluteus maximus, and hamstring, including the biceps femoris and semitendinosus. This underscored the importance of enhancing the synchronous activation ratio between the hamstrings and quadriceps to maintain dynamic knee stability, which is crucial in mitigating the risk of ACL injury (*Dedinsky et al., 2017*).

**Strengths and limitations of the study**

This study has several limitations stemming from the dataset and methodology. First, the small sample size (11 ACL injuries among 104 players) created significant class imbalance, limiting effective model training and external validation. Although VC dimension theory and cross-validation were used, lenient thresholds ($\epsilon = 0.1$, $\delta = 0.1$) may compromise robustness. Additionally, the number of injury cases was limited. This made it difficult to quantitatively assess the impact of Gaussian noise and SMOTE augmentation on feature distributions. Nevertheless, the modeling process was guided by iterative optimization, with model configurations were selected based on comprehensive performance metrics to ensure reliability within the constraints of the dataset. These findings underscored the need for larger and more diverse datasets to enhance the model's reliability and external validity. Furthermore, feature selection relied on VIF filtering to control multicollinearity. While this ensured stable model coefficients, it might have discarded informative predictors that were correlated, underscoring the need for more integrative feature selection approaches.

The participants in this study cohort was relatively homogeneous, consisting exclusively of experienced adult male basketball players, which limits the generalizability of the findings to broader populations. The experimental task employed an unanticipated side-cutting maneuver to enhance safety and reproducibility; however, it may not fully replicate the high-speed, sport-specific cutting actions typically associated with ACL injuries, thereby limiting the translational value of the findings. Furthermore, the scope of assessment was confined to biomechanical and neuromuscular variables, without incorporating physiological markers such as $VO_2$ max or blood lactate levels, which may influence the risk of ACL injury under fatigue (*Zago et al., 2021*). EMG data were collected only from the dominant leg, precluding the evaluation of bilateral neuromuscular asymmetries or compensatory mechanisms. Additionally, excluding the semitendinosus from EMG assessment may limit the evaluation of its role in knee stabilization. To improve predictive accuracy and applicability, future research should include diverse participant populations, bilateral EMG assessments, and relevant physiological indicators.

## CONCLUSIONS

The study identified key risk factors for ACL injury in basketball players using a machine learning–based predictive model. The most influential factors include biomechanical patterns during unanticipated side-cutting maneuvers, excessive activation of the vastus lateralis (with higher activation amplitudes correlating with increased injury susceptibility), lack of dynamic balance, and excessive training volume. The RF algorithm exhibited strong predictive accuracy, with feature importance analyses offering actionable insights for

targeted prevention strategies. These results reinforce the established role of biomechanical and neuromuscular mechanisms in ACL injury risk while advancing the field through data-driven approaches that improve predictive precision.

Based on these findings, preventive training should prioritize improved neuromuscular control of the knee and hip joints, increased knee flexion angles during landing, and optimized lower-limb kinematics to reduce internal tibial rotation torque. Moreover, enhanced activation of the vastus medialis and hamstring muscles is recommended to mitigate imbalances caused by overactivation of the vastus lateralis. Finally, we recommend that the training load should be less than 15 h per week with regular assessment of dynamic balance and squat jump performance are necessary to reduce the risk of ACL injury.

### Funding
The authors received no funding for this work.

### Competing Interests
The authors declare there are no competing interests.

### Author Contributions

- Longfei Guo conceived and designed the experiments, prepared figures and/or tables, and approved the final draft.
- Zhilei Cui performed the experiments, prepared figures and/or tables, and approved the final draft.
- Wei Ping Loh analyzed the data, prepared figures and/or tables, and approved the final draft.
- Shazlin Shaharudin analyzed the data, prepared figures and/or tables, authored or reviewed drafts of the article, and approved the final draft.

### Human Ethics
The following information was supplied relating to ethical approvals (i.e., approving body and any reference numbers):

The study protocol was approved by the Human Research Ethics Committee of Universiti Sains Malaysia (USM/JEPeM/22040199).

### Clinical Trial Ethics
The following information was supplied relating to ethical approvals (i.e., approving body and any reference numbers):

The study protocol was approved by the Human Research Ethics Committee of Universiti Sains Malaysia.

### Data Availability
The code and raw data are available in the Supplemental Files.

The raw data is available at Zenodo: Guo, L. (2025). Prediction of ACL injury [Data set]. Zenodo. https://doi.org/10.5281/zenodo.16930816.

## Clinical Trial Registration

The following information was supplied regarding Clinical Trial registration:
ISRCTN 18009799.

## Supplemental Information

Supplemental information for this article can be found online at http://dx.doi.org/10.7717/peerj.20141#supplemental-information.

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
