# Peer review of "Prediction of ACL injury incidence and analysis of key features in basketball players based on multi-algorithm models"

_PeerJ, doi:10.7717/peerj.20141_

## Round 0.1 · original submission · Major Revisions

Dear authors, thank you for your submission to PeerJ. After careful consideration of the 3 reviewers’ comments, the manuscript requires major revisions prior to being considered for publication. The key issues of concern include:

(1) insufficient clarity and consistency in the manuscript’s terminology, structure, and referencing

(2) they had significant concerns regarding the design and generalizability of the sidestep cutting task and lack of detail regarding data handling, trial selection, and missing data;

(3) there was limited integration of injury prevention insights into the discussion

(4) there was an underdeveloped literature review on machine learning applications in ACL injury prediction

(5) you had inadequate statistical reporting and interpretation, particularly regarding the model validation, feature contribution(s) and class imbalance. Further, the title and framing of the study may need to be reconsidered for better alignment with the sample characteristics and study limitations.

All three reviewers provided you with specific, constructive feedback to guide the necessary improvements required for your manuscript. Please address each point raised by the reviewers in a detailed revision (using track changes) and response to reviewers document. Thanks, A/Prof Mike Climstein

·

Basic reporting

The article is written with clear language, approaches a very significant phenomenon in the basketball sphere and seems to have used advanced tools to try to understand ACL injury occurrence.

The Introduction section lacks fluidity in the text, the phrases could be better puzzled together instead of being constantly separated by a period. The remainder of the article has a solid and fluent language.

There are some issues to be solved, which I will share with you further in this report.

Experimental design

1 - If the nutritional intake pre-task was controlled, the use of knee braces and sleeves should be mentioned in the text.
2 - Unfortunately fatigue can have a significant role on ACL injury occurrence, probably more than acute nutritional intake, this can influence the results in an unpredictable way.

Zhu A, Gao S, Huang L, Chen H, Zhang Q, Sun D, Gu Y. Effects of Fatigue and Unanticipated Factors on Knee Joint Biomechanics in Female Basketball Players during Cutting. Sensors (Basel). 2024 Jul 22;24(14):4759. doi: 10.3390/s24144759. PMID: 39066155; PMCID: PMC11280919.

Most right-handed people, kick with with the same leg, most of us, humans, have the same dominant hand and leg. However, when it comes to kicking, the dominant leg is generally more skilled, and since the non-dominant leg has more of a supporting role most people tend to be stronger with the non-dominant leg when it comes to jumping. That might hinder the results.

"A previous study identified muscle activation in the dominant leg as a primary contributor to
lower limb injuries [19]. Thus, this study exclusively measured the muscle activation of the
dominant leg, which was determined based on the participantís preferred kicking leg [20]."

Validity of the findings

The main problem with the findings is actually a naming problem for me. The validity of the study is much better if the article is themed:

Prediction of ACL injury recidive risk and analysis of key features in
basketball players based on multi-algorithm models

Instead of:

Prediction of ACL injury risk and analysis of key features in
basketball players based on multi-algorithm models

Because -> "Despite the studies contribution, this study had several limitations. Firstly, all participants
were experienced adult male basketball players who sustained ACL injury. This restricted the
generalisability of the findings to a broader population."


I do think that the difference of body weight profile between injured and non-injured could be explored too.


"Similar findings were reported by Stojmenovic et al. (2017), in which
those who trained for more than 10 hours weekly had 7.54 times higher odds of ACL injury than
those with shorter training durations [37]. Consistent with previous findings, this study
underscored the importance of identifying a unilateral safety threshold for weekly training loads
to mitigate injury risk of the athletes."

In this sentence are the odds calculated relative to the time of practice?

Are the odds growing because of the additional time of practice, or is indeed the presence of fatigue?

Additional comments

The change in the title of the article is, for me, essential. The rest of the problems that arose I think there are ways to mitigate them.

Reviewer 2 ·

Basic reporting

Please see the additional comments.

Experimental design

Please see the additional comments.

Validity of the findings

Please see the additional comments.

Additional comments

1. The overall language of the paper is professional; however, some sentences are overly long, which may affect readability. For instance, certain sentences in the abstract and methods sections have complex structures. It is recommended to simplify these sentences to improve clarity. Consider having a native speaker review and refine the language.
2. The introduction discusses research related to ACL injuries but provides limited coverage of machine learning (ML) applications in ACL injury prediction. It is recommended to include more references on ML algorithms used in sports injury prediction to highlight the novelty of this study. To provide more effective evidence, the authors may consider referring to the following relevant studies: Accurately and effectively predict the ACL force: Utilizing biomechanical landing pattern before and after-fatigue (https://doi.org/10.1016/j.cmpb.2023.107761). In addition, the introduction section lacks sufficient relevant references, which hardly explains the necessity and innovation of the study. Only 13 relevant references need to be added and only 5 relevant studies are recent in the last five years. Based on this, in addition to referencing the above-mentioned relevant studies, the following recent ACL injury literature could also be considered: New insights optimize landing strategies to reduce lower limb injury risk (https://doi.org/10.34133/cbsystems.0126). Next, as mentioned above in related studies, it is necessary to consider the reduction of ACL modeling to nonlinear short-term viscoelastic properties, which has been shown to restore ACL mechanical properties to a greater extent.

3. Figure 3 (AUC-ROC curves) should include a clearer legend to help readers better understand the performance comparison of different models. Figure 4 (Feature Importance Ranking) should provide explicit feature labels to reduce interpretation difficulties. Table 1 should include p-values and effect sizes to enhance the statistical rigor of data interpretation.
4. The paper mentions the availability of raw data but does not specify how the data can be accessed. It is recommended to provide a data repository link (e.g., a public data archive) and describe the data format and variable definitions to improve research reproducibility.
5. The study is based on 104 basketball players, with only 11 ACL injury cases. This severe class imbalance may lead to insufficient model training, affecting its generalization ability. It is suggested to discuss the impact of sample size on model performance in the discussion section or consider expanding the dataset in future research.
6. The study uses Gaussian noise and SMOTE for data augmentation but does not detail how augmentation affects feature distribution. It is recommended to provide experimental results comparing statistical characteristics before and after augmentation to ensure data augmentation does not introduce distortions.
7. The study applies the Variance Inflation Factor (VIF) to remove collinear features, but it does not clarify whether the final selected feature subset is appropriate for all models. It is suggested to explore how different feature selection methods impact model performance and provide supporting experimental results.
8. The study primarily compares model performance using AUC-ROC but does not conduct statistical tests. For example, DeLong’s test can be used to determine whether the differences in AUC between models are statistically significant.
9. Although SHAP is used to interpret the model, there is insufficient discussion on how each feature contributes to ACL injury risk. It is recommended to include biomechanical explanations of key features in the discussion section and compare them with existing literature to enhance the study’s clinical relevance.
10. The study’s conclusion is somewhat general. It is recommended to provide practical application insights, such as how the findings can be implemented in basketball training programs to prevent ACL injuries and potential future intervention strategies.

·

Basic reporting

The introduction shed light on the background and rationale of the study. Some references are missing or incorrectly used in this section (see general comments). The manuscript is generally well written, although some sentences could be improved (see general comments). Comments about the figures, tables and raw data are also included in the general comments part.

Experimental design

The research question is clear and scientifically addressed. However, the valuable insights to improve injury prevention that are intended to reach are not addressed sufficiently in the discussion section. The methods and results are not fully described to be able to replicate the study, see also my general comments.

Validity of the findings

One of my main concerns is the applied sidestep cutting task. The task used in this study used a 7m approach run, followed by a double leg stop after which a light appeared that indicated the direction to make the cut. So far I understand, the double leg stop prevents a real on-speed sidestep cut as currently often applied in scientific studies, prevention programs and, above all, what happened while rupturing ACL. This also influences the generalizability of the current findings and comparisons with previously performed studies.

Additional comments

I commend the authors for their extensive data set of 104 participants that have been tested twice (with one year in between). The authors applied several ML models to predict ACL injury based on several variables of different assessments (physical function, biomechanics of several tasks, balance, muscle strength). The current manuscript has potential for publication although several main comments prevent this at this stage.
My main concern refers to the applied task as stated in ‘3. Validity of the findings’.
Secondly, the authors are advised to be more clear and precise about the trials which are (or not) used for the analysis and the missing data. Some numbers are provided, but explanation about causes is not given, even as the % of i.e., the outliers to place it in perspective.
The manuscript could be improved even more by being consistent regarding terminology of left/right, dominant/non-dominant, supporting/non-supporting leg. It is really hard to interpret the findings in the current form. Also, please use one word for athletes/players/subjects etc., and maneuvers/task/moments etc.
Lastly, the current and forgotten references make the manuscript less scientific, this warrants against acceptance of the manuscript. There are multiple mentioned references which are not used well (i.e., do not provide information regarding the point the authors have made in text) (see general points as well). As I could not check all references, there may be more incorrectly (interpreted) references.
Title: ‘Prediction of ACL injury risk’, as the authors used a 12-month follow-up to examine which variables were significant different between injured and non-injured players, I suggest leaving ‘risk’, as the results regarding incidence instead of risk.
Abstract;
‘Heightened’ compared to who/what?
‘accessing injury risk’, similar to comment about title, do you mean ‘injury risk’ or ‘injury’.
L4: This reference is not correct, I can’t find ‘basketball’ in the mentioned reference.
L4; typo ‘injury’ should be injuries.
L9; Please clarify why this study focused solely on male participants, the fact that male and female players have differences in anatomical structure and biomechanical characteristics is not evident enough to go for males instead of females.
L13; Please provide correct references for the LESS and CMAS. The [5] is not appropriate.
L15; Reference [6] is not well chosen here, please provide a better one.
L18; Reference [7] is stated after [8] and [9], please order them correctly.
L21; How are poor balance and joint laxity taking into account in the unanticipated sidestep cutting test. The reasoning from the other contributing factors to this test to overcome these limitations is not enough explained.
L34; It is not completely clear if this study [12] used machine learning to predict injury risk or injury incidence. This is important because the current study uses ML to predict injuries, and not risk. It is advised to explain this further. Moreover, the LESS can be used to assess injury risk, however it has its shortcomings as well because it is subjective and based on 2D video cameras while 3D movements are analyzed.
L41/44; Reference is missing.
L47-51; The objective of the study is clear; however I don’t see the valuable insights about improvements of injury prevention strategies coming back in the discussion section which is unfortunate.
L56; This sentence does not add anything to this section, please remove.
L66; Please specify if the 8 hours of exercise were constrained to only basketball or also other sports (strength training, or completely other sports as soccer)
L70; Please explain where the sample size was determined and why 11 participants withdrew.
L73-76; Please provide numbers with 1 decimal constantly.
L104; Please check the numbering of sections throughout the manuscript. Here you start with 2.2.2., where are 2.1 and 2.2.1?
L110; ‘Six valid trials’, what is meant with this? Does it mean that each participant had to perform six trials in the direction of the non-dominant leg? Or six trials in total, regardless of the direction?
L137; Why was an intensity of 50W chosen? Did the participants really get warm during the this 5 minutes? I can imagine that this is too easy and short to have a proper warm-up. Please explain in more detail the 5 minutes of stretching, why was this done?
L142; Typo, it should be ‘[23], and’ I assume.
L145; Extra line break
L176; this section could largely be improved by being more clear about which variables are examined in which phase ánd, even more remarkable, which sidestep cuts are included in the analysis (see also earlier comment on L110).
L183; Reference is missing.
L186; Are the moments internally or externally calculated?
L197; Why was this model chosen?
L198-202; Based on which reference were the cutoff frequencies chosen?
L227; the number of outliers seems to be quite high? Can you explain this in more detail? Probably refer to the total number of trials to place in perspective. Why were the outliers removed? It is part of the data, and it seems to be a high number of removals which influence your data.
L280; Please remove ‘and’ before ‘XGBoost’
L285; Please explain why the AUC-ROC was identified as the primary metric.
L294; Please elaborate and be more precise regarding the number of trials used for your analysis. If I understand correctly, six trials per participant were performed, counting to 6x104 = 624 trials in total? Half of them is to the dominant leg direction probably and is thus not used for the analysis. 595 lines are filled in the excel sheet ‘Raw data 2’ and 300 in ‘Raw data 3’…I don’t get where those numbers come from.
L297; Please provide to which groups of risk factors the five features with VIF values >5 belong.
L299; Please use RF from here on to point to Random Forest
L302; AUC or AUC-ROC
Table 1; please align all similarly (left or center). Please be consistent regarding upper- and lower-case letters and space between words. See also my comment about L324 regarding terminology of left and right leg.
Table 1; Why is the S-Wilk missing for the relative deadlift?
Table2; sensitivity of XGboost and RF are identical, is that correct? Please be concise regarding the decimals.
L317; Please explain what the Gini Importance method is.
L324; The smaller knee flexion angle corresponds to both legs if I understand figure 4 correctly. Please align text with figure. Moreover, I suggest making the direction of the variables clear in figure 4, i.e., ‘increased knee flexion moment’. To make it easier for the reader, I suggest replacing left and knee to other terms as left and knee depends on the direction the subject went. You may opt for ‘supporting’ and ‘non-supporting’ legs.
L352; This is an important statement, but how should this threshold be reached in the future? The authors give a starting point but do not give any suggestion on how to achieve this.
L355-361; This part is hard to digest as terminology is used interchangeably, left/right, dominant/non-dominant, please be consistent.
L365; ‘were’? Does it not exist anymore? Please give a reference for this as well.
L368; This seems counterintuitive; larger knee flexion moments while having smaller knee flexion angles. This could only be achieved when the backward GRF is extremely large, or the terminology of moments is not correctly used.
L370-371; I don’t get this sentence, please rephrase.
L384; Grammar of ‘was’, I suggest replacing this with ‘consists of’.
L384; Till here I can follow the line of reasoning. However, from here on the authors start writing about the mean training duration and how this differ between the two groups. For me, both groups seem to be experienced (4 and 5 years, also pointing to an earlier comment, one decimal would help to interpret those numbers as the ‘real’ difference could be from 0.2-1.8 years…) and thus I don’t get the point of lowering center of gravity, which also comes out of the blue regarding the preceding part, and so reducing ACL injury risk. This section could largely be improved by being more clear.
L406; This is not a grammatically correct sentence on its own.
L408; ‘Non-dominant left leg’ is really confusing.
L414; Study by Nedergaard does not correspond to ref [51] in reference list.
L437; Wording, how can SJ be a training method? A training method seems to me to consist of more exercises, and how you combine them in contrast to just one single test (SJ).
L443; Here the semitendinosus is mentioned as one of the key muscles for concentric contraction. Why were the gastrocnemius muscles used in this study?
L448; Although it is written with caution, I don’t get why the authors elaborated on the possibility of the relative deadlift as factor which may influence ACL injury risk. This needs further explanation, why was chosen for this variable and not for other variables that differ significantly between groups?
L456; ‘was’ should be ‘were’, or even better ‘are’.
L457; Please elaborate on the prevention strategies.
L460; This wording is confusing. I assume the authors mean that the only injury that was registered was an ACL injury, however it now reads as all participants sustained an ACL injury.
L465/473/475; Please provide references.
L483; Please provide directions of the activation differences of VL
L488;-490; Although it is good to mention this, this is new information in the conclusion which should have been mentioned earlier already.
L492; The currently provided valuable guidance is not comprehensive for practitioners such as coaches and trainers.
STROBE statement checklist; please check the line numbers provided, i.e., the numbers of generalizability and limitations do not correspond to the line numbers in text. Items 8 and 13b could currently not be checked, see also comment about it in text.

---

## Round 0.2 · Minor Revisions

Thank you for your submission to PeerJ. I note in your manuscript that you have written: "Should the paper be accepted, we will promptly deposit the data in the journal's designated repository and provide detailed documentation (including variable definitions and file formats) to ensure full reproducibility."

Please be advised that PeerJ requires that all underlying data be made available at the time of review, not after acceptance. This ensures that reviewers can fully evaluate the reproducibility and validity of your findings as part of the peer review process.

Accordingly, I kindly ask that you deposit your dataset in a suitable repository now and provide the relevant access details (e.g., DOI or stable link) within your revised submission. We look forward to receiving the updated submission including access to the supporting data so that the review process can continue.

---

## Round 0.3 · accepted · Accept

Thank you for your thorough revision of the manuscript. I can confirm that all reviewer comments have been addressed appropriately. As the previous reviewers were not re-invited, I have carefully assessed the revised version myself and am satisfied with the current, amended manuscript. I am therefore recommending your revised manuscript be accepted for publication in PeerJ. Thank you for supporting PeerJ and we look forward to receiving future manuscripts from your research team.